# Profluorescent Fluoroquinolone-Nitroxides for Investigating Antibiotic–Bacterial Interactions

**DOI:** 10.3390/antibiotics8010019

**Published:** 2019-03-04

**Authors:** Anthony D. Verderosa, Rabeb Dhouib, Kathryn E. Fairfull-Smith, Makrina Totsika

**Affiliations:** 1School of Chemistry, Physics and Mechanical Engineering, Queensland University of Technology, Brisbane, QLD 4001, Australia; anthony.verderosa@qut.edu.au; 2Institute of Health and Biomedical Innovation, School of Biomedical Sciences, Queensland University of Technology, Brisbane, QLD 4006, Australia; rabeb.dhouib@qut.edu.au

**Keywords:** antibiotics, fluorescent antibiotics, fluorescent probes, fluoroquinolones, profluorescent nitroxides, nitroxides

## Abstract

Fluorescent probes are widely used for imaging and measuring dynamic processes in living cells. Fluorescent antibiotics are valuable tools for examining antibiotic–bacterial interactions, antimicrobial resistance and elucidating antibiotic modes of action. Profluorescent nitroxides are ‘switch on’ fluorescent probes used to visualize and monitor intracellular free radical and redox processes in biological systems. Here, we have combined the inherent fluorescent and antimicrobial properties of the fluoroquinolone core structure with the fluorescence suppression capabilities of a nitroxide to produce the first example of a profluorescent fluoroquinolone-nitroxide probe. Fluoroquinolone-nitroxide (FN) **14** exhibited significant suppression of fluorescence (>36-fold), which could be restored via radical trapping (fluoroquinolone-methoxyamine **17**) or reduction to the corresponding hydroxylamine **20**. Importantly, FN **14** was able to enter both Gram-positive and Gram-negative bacterial cells, emitted a measurable fluorescence signal upon cell entry (switch on), and retained antibacterial activity. In conclusion, profluorescent nitroxide antibiotics offer a new powerful tool for visualizing antibiotic–bacterial interactions and researching intracellular chemical processes.

## 1. Introduction

Fluorescent antibiotics can provide valuable insight, often in real time, into interactions between antibiotics and bacterial/host cells. These innovative compounds have aided in elucidating antibiotic modes of action [1], assessing drug toxicity [2], facilitating diagnoses [3], and examining bacterial antimicrobial resistance [4,5]. The development of a fluorescent antibiotic probe generally utilizes one of two methodologies. A fluorophore, such as boron-dipyrromethene (BODIPY) [6], rhodamine [7] or dansyl [8], can be covalently linked to an existing antibiotic to produce a fluorophore–antibiotic conjugate (Figure 1A). Most fluorescent antibiotic probes are generated via this method; however, this can alter antibiotic binding to cellular targets, reduce antibiotic potency and modify pharmacokinetics [9]. Thus, an alternative approach is to utilize intrinsically fluorescent antibiotics, such as the fluoroquinolones [10] or anthraquinone glycosides [11], as this method negates the challenges associated with tethering a fluorophore to an existing antibiotic. While both are valid methods, neither produce a final product capable of monitoring both antibiotic–bacterial interactions and chemical processes involving free radical and redox reactions. This is important as the induction of free radical and redox processes within bacterial cells following treatment with antibiotics such as quinolones [12], aminoglycosides [13], rifampicin [14], and chloramphenicol [15] are well documented to play a central role in antimicrobial activity [16]. Consequently, monitoring these processes during treatment may provide additional insight into antimicrobial modes of action and resistance mechanisms. Profluorescent nitroxides (PFNs) are probes currently used to monitor free radical and redox processes in a variety of applications (Figure 1B).

A PFN consists of a nitroxide moiety covalently attached to a fluorophore and exhibits a substantial suppression of fluorescence (nitroxides are efficient quenchers of excited states) [18,19,20]. Once the free radical of the nitroxide is removed, either by radical trapping or through redox chemistry, fluorescence is restored. PFNs have been successfully utilized for a variety of applications, including the identification of radical-based reaction intermediates [21,22], assessing the oxidative capacity of pollution [23,24,25], investigating polymer degradation [26,27,28], detecting reactive oxygen species in biological systems [29,30,31,32,33], and most recently as bacteriological probes to monitor free radical and redox processes [34]. 

In this work, we sought to combine the properties of a PFN with a fluorescent antibiotic, in order to create a probe capable of monitoring both antibiotic–bacterial interactions and free radical and redox processes within bacteria. Herein, we present the design, synthesis, photophysical and biological evaluation of profluorescent fluoroquinolone–nitroxides based on the inherent antibacterial and fluorescent properties of the fluoroquinolone core structure.

## 2. Results and Discussion

In the design of our PFN antibiotic, we chose to exploit fluoroquinolone because it has been extensively used as an inherently fluorescent antibiotic [10,35,36] and has demonstrated activity against a variety of clinically important pathogens, including *Pseudomonas aeruginosa* [37], *Escherichia coli* [38], *Staphylococcus aureus* [39], and *Enterococcus faecalis* [40]. Furthermore, we have previously worked with the related molecule ciprofloxacin [41,42]. In order to utilize the fluoroquinolone core to generate a profluorescent nitroxide which would retain antimicrobial activity, we proposed incorporating the nitroxide moiety via an amine linker at the C-7 position (see compound **1** for numbering) of the fluoroquinolone. Functionalization at this position with amino-nitrogen heterocycles has been shown to facilitate DNA gyrase or topoisomerase IV binding and subsequently enhance potency against both Gram-positive and Gram-negative bacteria [43]. Hence, the amino-functionalized heterocyclic nitroxides **2**–**4** were selected to generate the desired profluorescent fluoroquinolone–nitroxides. The synthesis of fluoroquinolone–nitroxides (FNs) **15**–**17** began by utilizing a Buchwald–Hartwig amination of fluoroquinolone **1** with 4-amino-2,2,6,6-tetramethylpiperidin-1-yloxyl (amino-TEMPO) **2**, 5-amino-1,1,3,3-tetramethylisoindolin-2-yloxyl (amino-TMIO) **3** or 5-amino-1,1,3,3-tetraethylisoindolin-2-yloxyl (amino-TEIO) **4** using Pd(OAc)_2_ and BINAP in THF to afford the ethyl ester fluoroquinolone–nitroxides **8**–**10** in high yield (80–90%) (Scheme 1). Final deprotection of **8**–**10** via base mediated hydrolysis gave the desired FNs **14**–**16** in excellent yield (80–90%).

In addition to the FNs **14**–**16**, their corresponding fluoroquinolone–methoxyamine (FM) derivatives **17**–**19** were also synthesized, to examine the specific effect of the nitroxide moiety on fluoroquinolone fluorescence suppression. Furthermore, they could also serve as controls in assessing any antibacterial activity of the nitroxide moiety, and enable the intermediates to be well characterized by NMR spectroscopy (nitroxides are paramagnetic and typically display significantly broadened NMR spectroscopy signals). Utilizing Fenton conditions, the nitroxides **2**–**4** were reacted with hydrogen peroxide, iron (II) sulfate heptahydrate and DMSO to furnish methoxyamines **5**–**7** in moderate to excellent yield (77–95%). Subsequent amination, followed by base mediated deprotection, as described previously, afforded FMs **17**–**19** in high yield (81–87%) (Scheme 1).

With the new FNs **14**–**16** and their corresponding FMs **17**–**19** in hand, we proceeded to evaluate their photophysical properties. All FNs **14**–**16** and their corresponding FMs **17**–**19** displayed absorbance spectra, fluorescence spectra and extinction coefficients (Table 1) characteristic of fluoroquinolones [44]. However, FNs **15** and **16**, and FMs **18** and **19**, which contained aromatic isoindoline-based functionality, exhibited substantially reduced quantum yields (ՓF) (>10-fold lower) when compared to the compounds FN **14** and FM **17** (piperidine-based functionality). This reduction in fluorescence potentially arises from a disruption to the delocalized π-electron system of the quinolone core, by the amine linked aromatic system of the isoindoline. Consequently, these results indicate that the addition of an aromatic ring via an amine linkage to the fluoroquinolone core at the C-7 position negatively impacts the fluorescent intensity of the fluorophore.

A comparison of the fluorescence arising from solutions of FNs **14**–**16** and their corresponding FMs **17**–**19** in chloroform identified a substantial fluorescence suppression in the presence of the nitroxide moieties (calculated from the ratio between the quantum yields of the corresponding methoxyamine and nitroxide conjugates) (Table 1). Fluorescence suppression was greatest in FNs **15** and **16** (67.5- and 75.0-fold, respectively), both of which bear the isoindoline core. FN **14** also demonstrated significant fluorescence suppression (27.7-fold), albeit lower than the other two nitroxide conjugates (FNs **15** and **16**). While these findings confirm that the physical properties of FNs **14**–**16** are suitable for profluorescent probe applications, these specific PFNs were designed for biological applications, and thus testing was repeated in an aqueous solution. Unfortunately, FNs **15**–**16** and their FM derivatives **18**–**19** displayed no measurable fluorescence in the aqueous solution, indicating water-induced fluorescence quenching. Thus, the photophysical properties of these compounds would not be optimal for aqueous biological applications (removal of the nitroxide would restore fluorescence, but no signal would be detectable due to water-induced fluorescence quenching). Conversely, FN **14** and FM **17** not only retained their fluorescence in the aqueous solution (Figure 2), but FM **17** actually produced a higher fluorescence quantum yield, which resulted in an improved suppression ratio in the aqueous solution compared to chloroform (Table 1). Subsequently, FN **14** and FM **17** were identified as possessing optimal photophysical properties for antibiotic–bacterial interaction studies. 

To assess the dynamic emission range of FN **14**, the reduction of FN **14** with an excess (1000 equivalents) of sodium ascorbate (Scheme 2), chosen due to its aqueous solubility, was conducted and monitored via fluorescence spectroscopy. Following treatment of FN **14** with sodium ascorbate in aqueous solution (distilled water, pH 7), reduction of the nitroxide (quenched species) to the corresponding hydroxylamine (fluorescent species), resulted in a steady increase in the fluorescence emission over time (measured every minute for 30 minutes) (Figure 3). Importantly, complete restoration of fluorescence was achieved (>99% switch on) compared to methoxyamine derivative **17** at the same concentration after approximately 30 minutes. This result suggests that the fluorescence quantum yield of FM **17** and the corresponding hydroxylamine derivative **20** of FN **14** are similar, and thus FM **17** is a true and accurate fluorescent control for FN **14**. Furthermore, the fact that the fluorescence of FN **14** was completely restored by removal of the free radical nitroxide highlights its profluorescent nature and demonstrates its potential value as a biological probe for visualizing free radical and redox processes. 

With the intended use of FN **14** and FM **17** in biological systems, we pondered the effect of pH on the fluorescence emission of FN **14** and FM **17**. Thus, we decided to examine the effect of pH on fluorescence intensity for FM **17** between the pH range of 0–14 (Figure 4). FM **17** demonstrated fluorescence intensity stability between pH 6–12, supporting its suitability as a bacteriological probe. Interestingly, the fluorescence intensity of FM **17** significantly increased (>2.5-fold) at lower pH (<6), indicating that protonation of the heteroatoms within the conjugated system of the fluorophore drastically increases the fluorescence output of the compound. Conversely, at higher pH (>12) the fluorescence of FM **17** was considerably reduced and eventually completely quenched (pH 14). While compounds FN **14** and FM **17** were not specifically designed as fluorescent pH probes, their pH-dependent fluorescence could certainly be exploited for this purpose in suitable applications. 

Following examination of the photophysical properties of FNs **14**–**16** and FMs **17**–**19**, we proceeded to determine whether these compounds retained antimicrobial activity. Our biological investigations were initiated with the screening of FNs **14**–**16** and FMs **17**–**19** in minimum inhibitory concentration (MIC) assays against the common Gram-negative pathogens *P. aeruginosa* and *E. coli*, and Gram-positive pathogens *S. aureus* and *E. faecalis* (Table 2). FNs **14** and **16** exhibited the highest activity against *S. aureus* (MIC ≤ 20 µM). Interestingly, their corresponding FMs **17** and **19** both demonstrated no activity against this species (MIC > 1200 µM), suggesting that the presence of the free radical nitroxide may mediate *S. aureus* antibacterial activity. This same trend was also observed against *E. faecalis* with FN **16** (MIC ≤ 310 µM) and the corresponding FM **19** (MIC > 600 µM). As this trend was not observed for the Gram-negative species tested (*P. aeruginosa* and *E. coli*), it suggests that this activity may be specific against Gram-positive bacteria. 

Furthermore, considering that nitroxides possess no inherent antibacterial activity (Appendix A, Appendix A), the difference between the nitroxide-containing conjugate and its corresponding methoxyamine derivative against Gram-positive species was surprising. While the activity of FNs **14** and **16** was highest against *S. aureus*, these two conjugates also exhibited Gram-negative antimicrobial activity, with FN **14** being most active against *E. coli* (MIC ≤ 100 µM) and FN **16** against *P. aeruginosa* (MIC ≤ 160 µM). The aqueous fluorescent properties of FN **14** combined with its antibacterial activity, suggesting this compound would be useful for monitoring antibiotic–bacterial interactions in both Gram-positive and Gram-negative bacteria.

As FN **14** and the corresponding FM **17** demonstrated optimal photophysical and biological properties, these compounds were subsequently evaluated for use as bacteriological probes. FN **14** and FM **17** were administered at two concentrations (150 µM and 600 µM) to *P. aeruginosa*, *E. coli*, *S. aureus*, and *E. faecalis* cells for 90 minutes, then visualized via fluorescence microscopy. FN **14** emitted bright fluorescence upon cell entry in all species tested (Figure 5B–E). However, FN **14** bacterial cell entry and fluorescence was found to be both concentration and species specific. When FN **14** was administered to *P. aeruginosa* at 150 µM, very few (~10%) cells fluoresced (Appendix A, Appendix A); however, when the concentration of FN **14** was increased to 600 µM, nearly all bacterial cells fluoresced (~90%) (Figure 5B). A similar pattern was observed with *E. coli* cells treated with FN **14** (Figure 5C). Interestingly, this concentration-dependent fluorescence output was not observed in Gram-positive bacteria (*S. aureus* and *E. faecalis*). In fact, when either *S. aureus* or *E. faecalis* were treated with FN **14** (150 µM), almost every bacterial cell emitted measurable fluorescence (~99%) (Figure 5D,E, respectively), suggesting that the process by which fluorescence is activated for FN **14** occurs more readily and/or more frequently in Gram-positive species. 

Intriguingly, despite FM **17** being a fluorescence activated derivative of FN **14**, and hence always fluorescent, it did not emit a measurable cell-associated fluorescence signal. Instead, fluorescence was only detected in the liquid medium, where FM **17** formed fluorescent aggregates (Figure 5G–J). The lack of any bacterial-associated fluorescence for FM **17** suggests that it either does not enter bacterial cells or its fluorescence is quenched intracellularly. The inability of FM **17**–**19** to translocate through the bacterial cell envelope could potentially explains their lack of antimicrobial activity against the Gram-positive pathogens *S. aureus* and *E. faecalis* (Table 1). However, this possibility would not explain their activity against Gram-negative pathogens *P. aeruginosa* and *E. coli*, where the potency of both the FNs **14**–**16** and their corresponding FMs **17**–**19** derivatives is conserved.

To test the hypothesis that fluorescence activation of FN **14** occurs intracellularly, we examined the fluorescence properties of FN **14** and FM **17** in medium only (no bacterial cells present). Here, we treated the medium with either 150 or 600 µM of FN **14** or FM **17** for 90 minutes. Our findings indicated that FN **14** in medium emits no measurable fluorescence (Figure 5A) (FN **14** also emitted no measurable fluorescence in PBS, LB, and MH). However, when bacterial cells were present in a medium containing FN **14**, they became highly fluorescent while the surrounding medium still exhibited no fluorescence (Figure 5B–E). Interestingly, in similar assays, FM **17** became immediately fluorescent in medium alone (Figure 5F) and could be seen to form small aggregates and crystals (Figure 5F–J) that were highly fluorescent. As FN **14** was not visible in medium but clearly visible inside bacterial cells, while FM **17** was only visible in medium, we can conclude that FN **14**’s fluorescence is activated via an intracellular process, and thus, FN **14** can function as a true intracellular bacteriological probe with the potential to simultaneously monitor antibiotic–bacterial interactions and intracellular free radical and redox processes.

Importantly, FN **14** exhibited a fluorescence signal which did not require background subtraction or correction for bacteria autofluorescence (bacteria autofluorescence was not detected under these conditions). Furthermore, while the experiments reported here utilized an excitation wavelength of ~365 nm, FN **14** was also efficiently excited by a 405 nm laser or a multiphoton laser set at 720 nm. Taken together these results demonstrate the utility of FN **14** and support its use as a potential live-cell imaging probe. 

## 3. Materials and Methods

### 3.1. General Methods

Synthetic reactions of an air-sensitive nature were carried out under an atmosphere of ultra-high purity argon. Anhydrous THF was obtained from the solvent purification system, Pure Solv^TM^ Micro, by Innovative Technologies. Anhydrous toluene was dried by storage over sodium wire. All other reagents were purchased from commercial suppliers and used without further purification. Ciprofloxacin, 7-chloro-1-cyclopropyl-6-fluoro-4-oxo-1,4-dihydroquinoline-3-carboxylic acid (Q-Acid), 4-carboxy-2,2,6,6-tetramethylpiperidin-1-yloxyl (CTEMPO), 2,2,6,6-tetramethylpiperidin-1-yloxyl (TEMPO), and 4-amino-2,2,6,6-tetramethylpiperidin-1-yloxyl (amino-TEMPO) **2** were purchased from Sigma-Aldrich Chemical Company. Ethyl 7-chloro-1-cyclopropyl-6-fluoro-4-oxo-1,4-dihydroquinoline-3-carboxylate **1** [45], 5-amino-1,1,3,3-tetramethylisoindolin-2-yloxyl (amino-TMIO) **3** [26], 5-nitro-1,1,3,3-tetraethylisoindolin-2-yloxyl **4** [46], 4-amino-1-methoxy-2,2,6,6-tetramethylpiperidine (amino-TEMPOMe) **5** [47], 5-amino-2-methoxy-1,1,3,3-tetramethylisoindoline (amino-TMIOMe) **6** [48], 5-nitro-1,1,3,3-tetraethylisoindolin-2-yloxyl (nitro-TEIO) [49], 1,1,3,3-tetramethylisoindolin-2-yloxyl (TMIO), [50] and 1,1,3,3-tetraethylisoindolin-2-yloxyl (TEIO) [51] were prepared in house by previously documented procedures. The analytical data obtained for each compound was consistent with that previously reported in the literature. All ^1^H NMR spectra were recorded at 600 MHz on a Bruker Avance 600 instrument. All ^13^C NMR spectra were recorded at 150 MHz on a Bruker Avance 600 instrument. Spectra were obtained in the following solvents: CDCl_3_ (reference peaks: ^1^H NMR: 7.26 ppm; ^13^C NMR: 77.19 ppm), CD_2_Cl_2_ (reference peaks: ^1^H NMR: 5.32 ppm; ^13^C NMR: 53.84 ppm), *d*_6_-DMSO (reference peaks: ^1^H NMR: 2.50 ppm; ^13^C NMR: 39.52 ppm) and CD_3_OD (reference peaks: ^1^H NMR: 3.31 ppm; ^13^C NMR: 49.00 ppm). All NMR experiments were performed at room temperature. Chemical shift values (δ) are reported in parts per million (ppm) for all ^1^H NMR and ^13^C NMR spectral assignments. ^1^H NMR spectroscopy multiplicities are reported as: s = singlet, br. s = broad singlet, d = doublet, dd = doublet of doublets, m = multiplet. Coupling constants are reported in Hz. All spectra are presented using MestReNova 9.0. High-resolution ESI mass spectra were obtained with a Thermo Fisher Scientific Orbitrap Elite mass spectrometer (Thermo Fisher Scientific, Waltham, MA, USA) or an Agilent Q-TOF LC high-resolution mass spectrometer, which utilized electrospray ionization in positive ion mode. Analytical HPLC was carried out on an Agilent Technologies HP 1100 Series HPLC system using an Agilent C18 column (250 mm × 4.6 mm × 5 μm) with a flow rate of 1 mL min^−1^. The purity of all final compounds was determined to be 95% or higher using HPLC analysis. EPR spectra were obtained with the aid of a miniscope MS 400 Magnettech EPR spectrometer. Column chromatography was performed using LC60A 40–63 Micron DAVISIL silica gel. Thin-layer chromatography (TLC) was performed on Merck Silica Gel 60 F254 plates. TLC plates were visualized under a UV lamp (254 nm) and/or by development with phosphomolybdic acid (PMA). Melting points were measured with a variable temperature apparatus by the capillary method and are uncorrected. Samples were separated by HPLC (Dionex Ultimate 3000) on a Phenomenex Luna C18 column (250 mm × 2.0 mm × 5 μm) held at 40 °C. Mobile phase A was 20% acetonitrile (ACN), and mobile phase B was 90% ACN, both containing 10 mM ammonium acetate, flowing at 0.2 mL min^−1^. The gradient commenced at 57% B for 3 minutes, increasing to 100% B over 7 minutes, and holding at 100% B for a further 5 minutes before returning to initial conditions for 5 minutes. Post-column, the eluent was split (~9:1) for both UV and MS detection. High-resolution mass spectra were acquired on an LTQ Orbitrap Elite mass spectrometer (Thermo Fisher Scientific, Bremen, Germany) equipped with a heated electrospray ionization source, operating in the positive ion mode with a mass resolution of 120,000 (FWHM at *m*/*z* 400). This method was used for nitro-TEIOMe with the following modifications: Isocratic run where mobile phase A was 20% ACN/80% water, containing 10 mM ammonium acetate, and mobile phase B was 100% MeOH, both flowing at 0.2 mL·min^−1^. All UV-Vis spectra were recorded on a single beam Varian Cary 50 UV-Vis spectrophotometer. Fluorescence measurements were performed on a Varian Cary 54 Eclipse fluorescence spectrophotometer equipped with a standard single-cell sample holder. Fluorescence microscopy was conducted on a Zeiss Axio Vert.A1 FL-LED equipped with filter sets, 49 (excitation: G 365 nm, beamsplitter: FT 395 nm, emission: BP 445/50 nm) used for FN 14 and FM 17. All microscopy experiments utilized a 100× oil immersion objective.

### 3.2. Synthesis of 5-Nitro-2-methoxy-1,1,3,3-tetraethylisoindoline (Nitro-TEIOMe)

Iron (II) sulfate heptahydrate (0.72 g, 2.58 mmol, 2.5 equiv) was added to a solution of nitro-TEIO (300 mg, 1.03 mmol, 1 equiv) in DMSO (10 mL). The mixture was cooled to 0 °C, and 35% aqueous hydrogen peroxide (276 µL, 4.12 mmol, 4 equiv) was added in a dropwise manner. The resulting mixture was stirred at 0 °C for 10 minutes and then at room temperature for an additional 1.5 hours. The reaction mixture was diluted with deionized water (40 mL) before being extracted with diethyl ether (3 × 20 mL). The combined organic extracts were washed with deionized water (200 mL) and dried over anhydrous sodium sulfate. The solvent was removed in vacuo to afford product nitro-TEIOMe as a light beige solid (285 mg, 0.93 mmol, 90%). ^1^H NMR (600 MHz, CDCl_3_): (*note the signals for the four ethyl groups were not observed on this NMR timescale) δ = 8.15 (dd, *J* = 8.3, 2.4 Hz, 1H, Ar-H), 7.98 (d, *J* = 2.4 Hz, 1H, Ar-H), 7.27 (dd, *J* = 8.3, 2.6 Hz, 1H, Ar-H), 3.80 (s, 3H, NOCH_3_). ^13^C NMR (150 MHz, CDCl_3_): δ = 152.8, 148.0, 147.2, 123.2, 122.6, 117.4, 67.4, 67.2, 65.8, 29.8, 24.8. HRMS (ESI): *m*/*z* calculated for C_7_H_27_N_2_O_3_ + H^+^ [M+H^+^]: 307.2022; found 307.2027. LC-MS Analysis: *R*_t_ = 10.9 min; area 99%. MP: 78.5–79.5 °C. 

### 3.3. Synthesis of 5-Amino-2-methoxy-1,1,3,3-tetraethylisoindoline (Amino-TEIOMe) ***7***

Palladium on carbon 10% wt. loading (87 mg, 0.082 mmol, 10 mol %) was added to a solution of nitro-TEIOMe (250 mg, 0.82 mmol, 1 equiv) in methanol (30 mL). The solution was placed in a Parr hydrogenator under an atmosphere of hydrogen gas (25 psi), with shaking, for 3.5 hours. The resulting solution was filtered through Celite, then acidified (pH 1) with aqueous hydrochloric acid (2 M) and extracted with diethyl ether (3 × 20 mL). The remaining aqueous solution was basified (pH 12) with sodium hydroxide (2 M) and extracted with diethyl ether (3 × 20 mL). The combined organic extracts were dried over anhydrous sodium sulfate, and the solvent was removed in vacuo to afford product **7** as a light yellow oil (215 mg, 0.78 mmol, 95%). ^1^H NMR (600 MHz, CDCl_3_): δ = 6.80 (d, *J* = 7.9 Hz, 1H, Ar-H), 6.56 (dd, *J* = 8.0, 2.3 Hz, 1H, Ar-H), 6.37 (d, *J* = 2.2 Hz, 1H, Ar-H), 3.67 (s, 3H, NOCH_3_), 2.03–1.93 (m, br, 4H, 2 × CH_2_), 1.72-1.67 (m, br, 4H, 2 × CH_2_), 0.92 (s, 6H, 2 × CH_3_), 0.78 (s, 6H, 2 × CH_3_). ^13^C NMR (150 MHz, CDCl_3_): δ = 144.5, 144.0, 133.2, 124.2, 114.0, 110.3, 72.8,72.4, 63.6, 30.1, 29.6, 9.6, 9.1. HRMS (ESI): *m*/*z* calculated for C_17_H_29_N_2_O + H^+^ [M+H^+^]: 277.2280; found 277.2282. LC-MS: *R*_t_ = 14.1 min; area 99%.

### 3.4. General Procedure for the Synthesis of Fluoroquinolone–Nitroxides ***14***–***16*** and Fluoroquinolone–Methoxyamines ***17***–***19***

Cesium carbonate (3 equiv), palladium acetate (6 mol %), BINAP (10 mol %), Q-Ester **1** (2 equiv) and the specific primary amine (1 equiv) were added to a Schlenk vessel under an atmosphere of argon. THF (60 mL), which had been degassed with argon, was then added. The vessel was sealed and heated at 65 °C for 72 hours. The reaction was allowed to cool to room temperature, and the solvent was removed via rotary evaporation. The resulting residue was washed three times with aqueous hydrochloric acid (2 M, 3 × 20 mL) and the combined filtrates were extracted with diethyl ether (3 × 10 mL). The aqueous phase was neutralized with saturated sodium carbonate and extracted with dichloromethane (3 × 50 mL). The combined extracts were dried over anhydrous sodium sulfate and the solvent removed in vacuo. Purification was achieved via column chromatography (SiO_2_, chloroform 98%, methanol 2%).

### 3.5. Ethyl 1-Cyclopropyl-6-fluoro-7-(2,2,6,6-tetramethyl-1-oxy-piperidine-4-yl)amino)-4-oxo-1,4-dihydroquinoline-3-carboxylate ***8***

Reagents: Cesium carbonate (860 mg, 2.64 mmol, 3 equiv), palladium acetate (12 mg, 0.053 mmol, 6 mol %), BINAP (55 mg, 0.088 mmol, 10 mol %), Q-Ester **1** (545 mg, 1.76 mmol, 2 equiv), amino-TEMPO **2** (150 mg, 0.88 mmol, 1 equiv). Product: Orange solid (340 mg, 0.77 mmol, 87%). ^1^H NMR (600 MHz, CDCl_3_): (*note compound is a free-radical, some signals appear broadened, and other signals are missing) δ = 8.38 (s, 1H, NCH=C), 7.95 (s, 1H, Ar-H), 7.09 (s, 1H, Ar-H), 4.27 (d, *J* = 6.8 Hz, 1H, OCH_2_CH_3_), 3.28 (s, 1H, C=CHNCH), 1.30 (t, *J* = 6.5 Hz, 3H, OCH_2_CH_3_), 1.10 (s, br, 4H, 2 × NCHCH_2_). ^13^C NMR (150 MHz, CDCl_3_): (*note compound is a free-radical, some signals appear broadened, and other signals are missing) δ = 172.2, 165.0, 147.0, 139.5, 138.5, 118.2, 111.0, 109.6, 94.1, 60.0, 33.9, 13.6, 9.2. MP: 214.5–215.4 °C. HRMS (ESI): *m*/*z* calculated for C_24_H_32_FN_3_O_4_ + H^+^ [M+H^+^]: 445.2377; found 445.2379. LC-MS: *R*_t_ = 5.0min; area 100%. EPR: g = 2.00009, a_N_ = 1.384 mT.

### 3.6. Ethyl 1-Cyclopropyl-6-fluoro-7-((1-methoxy-2,2,6,6-tetramethylpiperidine-4-yl)amino)-4-oxo-1,4-dihydroquinoline-3-carboxylate ***11***

Reagents: Cesium carbonate (860 mg, 2.64 mmol, 3 equiv), palladium acetate (12 mg, 0.053 mmol, 6 mol %), BINAP (55 mg, 0.088 mmol, 10 mol %), Q-Ester **1** (545 mg, 1.76 mmol, 2 equiv), amino-TEMPOMe 5 (165 mg, 0.88 mmol, 1 equiv). Product: Light yellow solid (354 mg, 0.77 mmol, 88%). ^1^H NMR (600 MHz, CDCl_3_): δ = 8.44 (s, 1H, NCH=C), 7.94 (d, *J* = 12.1 Hz, 1H, Ar-H), 6.94 (d, *J* = 7.0 Hz, 1H, Ar-H), 4.36 (q, *J* = 7.1 Hz, 3H, OCH_2_CH_3_), 4.31 (m, 1H, NH), 3.72 (m, 1H, C=CHNCH), 3.37 (m, 1H, NHCH), 1.96 (d, br, t, *J* = 12.2 Hz, 2H, NHCHCH_2_), 1.49 (t, *J* = 12.2 Hz, 2H, NHCHCH_2_), 1.39 (t, *J* = 7.1 Hz, 3H, OCH_2_CH_3_), 1.26–1.24 (s, 12H, 4 × CH_3_), 1.26–1.24 (m, 2H, NCHCH_2_), 1.14 (m, 2H, NCHCH_2_). ^13^C NMR (150 MHz, CDCl_3_): δ = 173.2, 166.2, 151.0, 148.6, 147.9, 104.2, 140.0, 139.1, 118.9, 111.6, 111.4, 110.4, 96.1, 65.7, 60.9, 60.0, 45.3, 44.0, 34.5, 33.0, 20.9, 14.6, 8.3. MP: 238.1–240.0 °C. HRMS (ESI): *m*/*z* calculated for C_25_H_35_FN_3_O_4_ + H^+^ [M+H^+^]: 460.2612; found 460.2611. LC-MS: *R*_t_ = 12.0 min; area 97%. 

### 3.7. Ethyl 1-Cyclopropyl-6-fluoro-7-((1,1,3,3-tetramethylisoindolin-2-yloxyl-5-yl)amino)-4-oxo-1,4-dihydroquinoline-3-carboxylate ***9***

Reagents: Cesium carbonate (860 mg, 2.64 mmol, 3 equiv), palladium acetate (12 mg, 0.053 mmol, 6 mol %), BINAP (55 mg, 0.088 mmol, 10 mol %), Q-Ester **1** (545 mg, 1.76 mmol, 2 equiv), amino-TMIO 3 (180 mg, 0.88 mmol, 1 equiv). Product: Yellow solid (378 mg, 0.79 mmol, 90%). ^1^H NMR (600 MHz, CDCl_3_): (*note compound is a free-radical, some signals appear broadened, and other signals are missing) δ = 8.51 (s, 1H, NCH=C), 8.17 (d, *J* = 9.2 Hz, 1H, Ar-H), 7.60 (s, br, 1H, Ar-H), 6.45 (s, br, 1H, Ar-NH), 4.41 (q, *J* = 6.9 Hz, 2H, OCH_2_CH_3_), 3.30 (s, br, 1H, C=CHNCH), 1.43 (t, *J* = 6.9 Hz, 3H, OCH_2_CH_3_), 1.30–0.93 (m, br, 4H, 2 × NCHCH_2_). ^13^C NMR (150 MHz, CDCl_3_): (*note compound is a free-radical, some signals appear broadened, and other signals are missing) δ = 172.7, 165.8, 147.9, 138.0, 121.4, 112.2, 112.0, 110.4, 99.5, 60.8, 34.6, 14.3, 8.6. MP: 216.5–217.7 °C. HRMS (ESI): *m*/*z* calculated for C_27_H_30_FN_3_O_4_ + H^+^ [M+H^+^]: 479.2220; found 479.2227. LC-MS: *R*_t_ = 5.7 min; area 98%. EPR: g = 2.00003, a_N_ = 1.433 mT.

### 3.8. Ethyl 1-Cyclopropyl-6-fluoro-7-((2-methoxy-1,1,3,3-tetramethylisoindoline-5-yl)amino)-4-oxo-1,4-dihydroquinoline-3-carboxylate ***12***

Reagents: Cesium carbonate (860 mg, 2.64 mmol, 3 equiv), palladium acetate (12 mg, 0.053 mmol, 6 mol %), BINAP (55 mg, 0.088 mmol, 10 mol %), Q-Ester **1** (545 mg, 1.76 mmol, 2 equiv), amino-TMIOMe 6 (182 mg, 0.88 mmol, 1 equiv). Product: Pale yellow solid (360 mg, 0.73 mmol, 83%). ^1^H NMR (600 MHz, CDCl_3_): δ = 8.48 (s, 1H, NCH=C), 8.11 (d, *J* = 11.8 Hz, 1H, Ar-H), 7.54 (d, *J* = 6.9 Hz, 2H, Ar-H), 7.20–7.11 (m, 2H, 2 × Ar-H), 7.07 (d, *J* = 1.8 Hz, 2H, Ar-H), 6.32 (d, *J* = 3.7 Hz, 1H, Ar-NH), 4.38 (q, *J* = 7.1 Hz, 2H, OCH_2_CH_3_), 3.79 (s, 3H, NOCH_3_), 3.25 (tt, *J* = 7.1, 4.0 Hz, 1H, C=CHNCH), 1.45–1.39 (s, br, 12H, 4 × CH_3_), 1.40 (t, *J* = 7.1 Hz, 3H, OCH_2_CH_3_), 1.19–1.12 (m, 2H, NCHCH_2_), 1.10–1.05 (m, 2H, NCHCH_2_). ^13^C NMR (150 MHz, CDCl_3_): δ = 173.2, 166.2, 151.0, 149.4, 148.2, 147.1, 138.7, 138.6, 138.1, 123.0, 121.4, 121.0, 121.9, 115.4, 112.4, 112.3, 110.5, 99.0, 67.2, 67.1, 65.7, 61.0, 34.6, 14.6, 8.3. MP: 242.8–243.6 °C. HRMS (ESI): *m*/*z* calculated for C_28_H_33_FN_3_O_4_ + H^+^ [M+H^+^]: 494.2455; found 494.2456. LC-MS: *R*_t_ = 13.7 min; area 98%. 

### 3.9. Ethyl 1-Cyclopropyl-6-fluoro-7-((1,1,3,3-tetraethylisoindolin-2-yloxyl-5-yl)amino)-4-oxo-1,4-dihydroquinoline-3-carboxylate ***10***

Reagents: Cesium carbonate (860 mg, 2.64 mmol, 3 equiv), palladium acetate (12 mg, 0.053 mmol, 6 mol %), BINAP (55 mg, 0.088 mmol, 10 mol %), Q-Ester **1** (545 mg, 1.76 mmol, 2 equiv), amino-TEIO 4 (230 mg, 0.88 mmol, 1 equiv). Product: Yellow solid (376 mg, 0.70 mmol, 80%). ^1^H NMR (600 MHz, CDCl_3_): (*note compound is a free-radical, some signals appear broadened, and other signals are missing) δ = 8.50 (s, 1H, NCH=C), 8.16 (d, *J* = 9.6 Hz, 1H, Ar-H), 7.53 (s, br, 1H, Ar-H), 6.39 (s, 1H, Ar-NH), 4.40 (q, *J* = 7.0 Hz, 2H, OCH_2_CH_3_), 3.27 (s, 1H, C=CHNCH), 1.42 (t, *J* = 7.0 Hz, 3H, OCH_2_CH_3_), 1.10 (s, br, 4H, 2 × NCHCH_2_). ^13^C NMR (150 MHz, CDCl_3_): (*note compound is a free-radical, some signals appear broadened, and other signals are missing) δ = 172.8, 165.9, 148.0, 138.2, 121.3, 112.3, 110.5, 99.3, 60.9, 34.7, 14.4, 8.5. MP: 178.5–180.0 °C (decomposed). HRMS (ESI): *m*/*z* calculated for C_31_H_38_FN_3_O_4_ + H^+^ [M+H^+^]: 534.2849; found 534.2850. LC-MS: *R*_t_ = 12.1 min; area 97%. EPR: g = 2.00012, a_N_ = 1.394 mT.

### 3.10. Ethyl 1-Cyclopropyl-6-fluoro-7-((2-methoxy-1,1,3,3-tetraethylisoindoline-5-yl)amino)-4-oxo-1,4-dihydroquinoline-3-carboxylate ***13***

Reagents: Cesium carbonate (860 mg, 2.64 mmol, 3 equiv), palladium acetate (12 mg, 0.053 mmol, 6 mol %), BINAP (55 mg, 0.088 mmol, 10 mol %), Q-Ester **1** (545 mg, 1.76 mmol, 2 equiv), amino-TEIOMe 7 (243 mg, 0.88 mmol, 1 equiv). Product: Light yellow solid (406 mg, 0.74 mmol, 84%). ^1^H NMR (600 MHz, CDCl_3_): δ = 8.47 (s, 1H, NCH=C), 8.12 (d, *J* = 11.8 Hz, 1H, Ar-H), 7.41 (d, *J* = 7.0 Hz, 1H, Ar-H), 7.17 (dd, *J* = 8.1, 2.1 Hz, 1H, Ar-H), 7.08 (d, *J* = 8.1 Hz, 1H, Ar-H), 6.96 (d, *J* = 2.1 Hz, 1H, Ar-H), 6.30 (d, *J* = 3.7 Hz, 1H, Ar-NH), 4.38 (q, *J* = 7.1 Hz, 2H, OCH_2_CH_3_), 3.70 (s, 3H, NOCH_3_), 3.22 (td, *J* = 7.1, 3.6 Hz, 1H, C=CHNCH), 2.17–1.89 (m, br, 4H, 2 × CCH_2_CH_3_), 1.89–1.65 (m, br, 4H, 2 × CCH_2_CH_3_), 1.40 (t, *J* = 7.1 Hz, 3H, OCH_2_CH_3_), 1.32–1.19 (m, br, 2H, NCHCH_2_), 1.11–1.04 (m, br, 2H, NCHCH_2_), 1.04–0.91 (s, br, 6H, 3 × CCH_2_CH_3_), 0.87–0.70 (s, br, 6H, 3 × CCH_2_CH_3_). ^13^C NMR (150 MHz, CDCl_3_): δ = 173.2, 166.3, 151.0, 149.4, 148.1, 144.8, 139.8, 138.8, 137.7, 124.8, 121.3, 117.7, 112.4, 112.3, 110.5, 98.9, 72.9, 72.8, 63.7, 51.0, 34.6, 29.5, 14.6, 9.2, 8.2. MP: 214.8–217.6 °C. HRMS (ESI): *m*/*z* calculated for C_32_H_41_FN_3_O_4_ + H^+^ [M+H^+^]: 550.3081; found 550.3081. LC-MS: *R*_t_ = 19.6 min; area 96%.

### 3.11. General Procedure for the Synthesis of FNs ***14***–***19*** via Base Mediated Ester Hydrolysis

Aqueous sodium hydroxide (2 M, 7 equiv) was added to a solution of the specific ethyl ester (1 equiv) in HPLC grade methanol (50 mL), and the resulting solution was stirred at 50 °C for 5 hours. The reaction mixture was cooled to room temperature and diluted with deionized water (50 mL). The pH was adjusted to approximately 6 using aqueous hydrochloric acid (2 M) and the mixture extracted with dichloromethane (3 × 20 mL). The combined organic extracts were dried over anhydrous sodium sulfate, and the solvent was removed in vacuo. Purification was achieved via column chromatography (SiO_2_, chloroform 98%, methanol 2%).

### 3.12. 1-Cyclopropyl-6-fluoro-7-(2,2,6,6-tetramethyl-1-oxy-piperidine-4-yl)amino)-4-oxo-1,4-dihydroquinoline-3-carboxylic acid ***14***

Reagents: **8** (49 mg, 0.11 mmol, 1 equiv), aqueous sodium hydroxide (2 M, 0.39 mL, 0.77 mmol, 7 equiv) and HPLC grade methanol (10 mL). Product: Orange powdery solid (41 mg, 0.10 mmol, 90%). ^1^H NMR (600 MHz, CDCl_3_): (*note compound is a free-radical, some signals appear broadened, and other signals are missing) δ = 15.20 (s, 1H, COOH), 8.74 (s, 1H, NCH=C), 8.05 (s, 1H, Ar-H), 7.78–7.29 (s, br, 1H, Ar-H), 4.86–4.20 (s, br, 1H Ar-NH), 3.52 (s, 1H, C=CHNCH), 1.39–1.09 (m, br, 4H, 2 × NCHCH_2_). ^13^C NMR (150 MHz, CDCl_3_): (*note compound is a free-radical, some signals appear broadened, and other signals are missing) δ = 190.7, 177.0, 168.9, 167.2, 147.2, 140.6, 110.9, 108.1, 35.5, 22.2. MP: 253.8–255.0 °C. HRMS (ESI): *m*/*z* calculated for C_22_H_28_FN_3_O_4_ + H^+^ [M+H^+^]: 417.2064; found 417.2068. LC-MS: *R*_t_ = 4.98 min; area 100%. HPLC analysis: Retention time = 2.174 min; peak area = 99%; eluent A, Acetonitrile; eluent B, H_2_O (TFA 0.1%); isocratic (99:1) over 20 min with a flow rate of 1 mL min^−1^ and detected at 254 nm; C18 column; column temperature, rt. EPR: g = 2.00029, a_N_ = 1.547 mT.

### 3.13. 1-Cyclopropyl-6-fluoro-7-((1-methoxy-2,2,6,6-tetramethylpiperidine-4-yl)amino)-4-oxo-1,4-dihydroquinoline-3-carboxylic acid ***17***

Reagents: **11** (50 mg, 0.11 mmol, 1 equiv), aqueous sodium hydroxide (2 M, 0.39 mL, 0.77 mmol, 7 equiv) and HPLC grade methanol (10 mL). Product: White powdery solid (43 mg, 0.10 mmol, 87%). ^1^H NMR (600 MHz, CDCl_3_): δ = 15.31 (s, 1H, COOH), 8.73 (s, 1H, NCH=C), 7.97 (d, *J* = 11.6 Hz, 1H, Ar-H), 7.06 (d, *J* = 6.9 Hz, 1H, Ar-H), 4.56 (s, 1H, Ar-NH), 3.80–3.72 (m, 1H, C=CHNCH), 3.65 (s, 3H, NOCH_3_), 3.49 (tt, *J* = 7.2, 4.1 Hz, 1H, NHCH), 1.99 (d, *J* = 13.3 Hz, 2H, NHCHCH_2_), 1.53 (d, *J* = 12.5 Hz, 2H, NHCHCH_2_), 1.36–1.31 (m, 2H, NCHCH_2_), 1.28 (s, 12H, 4 × CH_3_), 1.23–1.15 (m, 2H, NCHCH_2_). ^13^C NMR (150 MHz, CDCl_3_): δ = 177.1, 167.5, 147.2, 104.4, 137.4, 117.3, 115.9, 110.7, 110.5, 108.1, 104.6, 100.1, 96.0, 65.8, 60.0, 45.1, 44.2, 35.3, 33.0, 21.0, 8.4. MP: 283.7–284.9 °C. HRMS (ESI): *m*/*z* calculated for C_23_H_31_FN_3_O_4_ + H^+^ [M+H^+^]: 432.2299; found 432.2299. LC-MS: *R*_t_ = 12.19 min; area 96%. HPLC analysis: Retention time = 2.612 min; peak area = 95%; eluent A, Acetonitrile; eluent B, H_2_O (TFA 0.1%); isocratic (99:1) over 20 min with a flow rate of 1 mL min^−1^ and detected at 254 nm; C18 column; column temperature, rt.

### 3.14. 1-Cyclopropyl-6-fluoro-7-((1,1,3,3-tetramethylisoindolin-2-yloxyl-5-yl)amino)-4-oxo-1,4-dihydroquinoline-3-carboxylic acid ***15***

Reagents: **9** (52 mg, 0.11 mmol, 1 equiv), aqueous sodium hydroxide (2 M, 0.39 mL, 0.77 mmol, 7 equiv) and HPLC grade methanol (10 mL). Product: Yellow powdery solid (40 mg, 0.09 mmol, 83%). ^1^H NMR (600 MHz, CDCl_3_): (*note compound is a free-radical, some signals appear broadened, and other signals are missing) δ = 15.07 (s, 1H, COOH), 8.75 (s, 1H, NCH=C), 8.15 (d, *J* = 10.1 Hz, 1H, Ar-H), 7.67 (s, 1H, Ar-H), 6.66 (s, 1H, Ar-NH), 3.39 (s, 1H, NHCH), 1.40 – 1.22 (m, br, 4H, 2 × NCHCH_2_). ^13^C NMR (150 MHz, CDCl_3_): (*note compound is a free-radical, some signals appear broadened, and other signals are missing) δ = 176.8, 166.9, 147.3, 139.3, 121.9, 118.3, 111.2, 108.1, 99.2, 35.5, 29.6, 8.8. MP: 253.3–254.6 °C. HRMS (ESI): *m*/*z* calculated for C_25_H_26_N_3_O_4_ + H^+^ [M+H^+^]: 451.1907; found 451.1913. LC-MS: *R*_t_ = 5.49 min; area 98%. HPLC analysis: Retention time = 2.225 min; peak area = 96%; eluent A, Acetonitrile; eluent B, H_2_O (TFA 0.1%); isocratic (99:1) over 20 min with a flow rate of 1 mL min^−1^ and detected at 254 nm; C18 column; column temperature, rt. EPR: g = 2.00006, a_N_ = 1.433 mT.

### 3.15. 1-Cyclopropyl-6-fluoro-7-((2-methoxy-1,1,3,3-tetramethylisoindoline-5-yl)amino)-4-oxo-1,4-dihydroquinoline-3-carboxylic acid ***18***

Reagents: **12** (54 mg, 0.11 mmol, 1 equiv), aqueous sodium hydroxide (2 M, 0.39 mL, 0.77 mmol, 7 equiv) and HPLC grade methanol (10 mL). Product: Pale yellow powdery solid (42 mg, 0.09 mmol, 82%). ^1^H NMR (600 MHz, CDCl_3_): δ = 15.17 (s. 1H, COOH), 8.71 (s, 1H, NCH=C), 8.08 (d, *J* = 11.5 Hz, 1H, Ar-H), 7.60 (d, *J* = 7.0 Hz, 1H, Ar-H), 7.23–7.13 (m, 2H, Ar-H), 7.09 (d, *J* = 1.8 Hz, 1H, Ar-H), 6.51 (d, *J* = 3.8 Hz, 1H, Ar-NH), 3.80 (s, 3H, NOCH_3_), 3.36 (tt, *J* = 7.1, 4.0 Hz, 1H, NHCH), 1.46 (s, br, 12H, 4 × CH_3_), 1.24–1.17 (m, 2H, NCHCH_2_), 1.17–1.09 (m, 2H, NCHCH_2_). ^13^C NMR (150 MHz, CDCl_3_): δ = 177.1, 167.3, 151.3, 149.7, 147.5, 147.2, 143.1, 140.0, 139.9, 138.6, 137.7, 123.2, 123.0, 122.1, 121.1, 117.7, 116.1, 115.3, 111.4, 111.2, 108.1, 99.0, 98.8, 67.2, 67.1, 65.7, 52.2, 35.4, 34.7, 29.8, 24.9, 8.3. MP: 293.7–295.3 °C. HRMS (ESI): *m*/*z* calculated for C_26_H_29_N_3_O_4_ + H^+^ [M+H^+^]: 466.2142; found 466.2148. LC-MS: *R*_t_ = 13.64 min; area 98%. HPLC analysis: Retention time = 3.243 min; peak area = 98%; eluent A, Acetonitrile; eluent B, H_2_O (TFA 0.1%); isocratic (99:1) over 20 min with a flow rate of 1 mL min^−1^ and detected at 254 nm; C18 column; column temperature, rt.

### 3.16. 1-Cyclopropyl-6-fluoro-7-((1,1,3,3-tetraethylisoindolin-2-yloxyl-5-yl)amino)-4-oxo-1,4-dihydroquinoline-3-carboxylic acid ***16***

Reagents: **10** (59 mg, 0.11 mmol, 1 equiv), aqueous sodium hydroxide (2 M, 0.39 mL, 0.77 mmol, 7 equiv) and HPLC grade methanol (10 mL). Product: Pale yellow powdery solid (45 mg, 0.09 mmol, 80%). ^1^H NMR (600 MHz, CDCl_3_): (*note compound is a free-radical, some signals appear broadened, and other signals are missing) δ = 15.08 (s. 1H, COOH), 8.73 (s, 1H, NCH=C), 8.13 (d, *J* = 9.0 Hz, 1H, Ar-H), 7.59 (s, 1H, Ar-H), 6.60 (s, 1H, Ar-NH), 3.38 (s, 1H, NHCH), 1.39–1.22 (m, br, 2H, NCHCH_2_), 1.22–1.08 (m, br, 2H, NCHCH_2_). ^13^C NMR (150 MHz, CDCl_3_): (*note compound is a free-radical, some signals appear broadened, and other signals are missing) δ = 176.8, 166.9, 147.3, 139.4, 118.0, 111.3, 111.1, 108.0, 98.9, 35.4, 29.6, 8.5. MP: 245.1–246.2 °C. HRMS (ESI): *m*/*z* calculated for C_29_H_34_N_3_O_4_ + H^+^ [M+H^+^]: 507.2533; found 507.2537. LC-MS: *R*_t_ = 12.27 min; area 98%. HPLC analysis: Retention time = 3.140 min; peak area = 96%; eluent A, Acetonitrile; eluent B, H_2_O (TFA 0.1%); isocratic (80:20) over 20 min with a flow rate of 1 mL min^−1^ and detected at 254 nm; C18 column; column temperature, rt. EPR: g = 2.00009, a_N_ = 1.379 mT.

### 3.17. 1-Cyclopropyl-6-fluoro-7-((2-methoxy-1,1,3,3-tetraethylisoindoline-5-yl)amino)-4-oxo-1,4-dihydroquinoline-3-carboxylic acid ***19***

Reagents: **13** (60 mg, 0.11 mmol, 1 equiv), aqueous sodium hydroxide (2 M, 0.39 mL, 0.77 mmol, 7 equiv) and HPLC grade methanol (10 mL). Product: Pale yellow powdery solid (47 mg, 0.09 mmol, 81%). ^1^H NMR (600 MHz, CDCl_3_): δ = 15.19 (s, 1H, COOH), 8.71 (s, 1H, NCH=C), 8.09 (d, *J* = 11.4 Hz, 1H, Ar-H), 7.45 (d, *J* = 7.0 Hz, 1H, Ar-H), 7.20 (dd, *J* = 8.0, 2.1 Hz, 1H, Ar-H), 7.12 (d, *J* = 8.0 Hz, 1H, Ar-H), 7.12 (d, *J* = 8.0 Hz, 1H, Ar-H), 6.49 (d, *J* = 3.8 Hz, 1H, Ar-NH), 3.70 (s, 3H, NOCH_3_), 3.32 (ddd, *J* = 10.7, 7.1, 4.1 Hz, 1H, NHCH), 2.19–1.92 (m, br, 4H, CCH_2_CH_3_), 1.88–1.66 (m, br, 4H, CCH_2_CH_3_), 1.17–1.09 (m, 2H, NCHCH_2_), 1.09–1.04 (m, 2H, NCHCH_2_), 1.03–0.88 (s, br, 6H, 2 × CCH_2_CH_3_), 0.88–0.70 (s, br, 6H, 2 × CCH_2_CH_3_). ^13^C NMR (150 MHz, CDCl_3_): δ = 177.1, 167.4, 151.3, 149.6, 147.3, 145.1, 140.9, 140.5, 140.4, 139.9, 124.9, 122.0, 118.6, 117.6, 111.4, 111.2, 108.1, 98.6, 73.0, 72.9, 63.7, 35.4, 30.1, 29.6, 9.6, 9.1, 8.3. MP: 270.5–272.0 °C. HRMS (ESI): *m*/*z* calculated for C_30_H_37_N_3_O_4_ + H^+^ [M+H^+^]: 522.2768; found 522.2771. LC-MS: *R*_t_ = 19.13 min; area 97%. HPLC analysis: Retention time = 7.328 min; peak area = 95%; eluent A, Acetonitrile; eluent B, H_2_O (TFA 0.1%); isocratic (80:20) over 20 min with a flow rate of 1 mL min^−1^ and detected at 254 nm; C18 column; column temperature, rt.

### 3.18. Fluorescence Quantum Yield and Extinction Coefficient Calculations

Quantum yield efficiencies of fluorescence for compounds **14**–**19** were obtained from measurements at five different concentrations in water, ethanol, or chloroform using the following equation:*Ф*_F sample_ = *Ф*_F standard_ × (*A*_standard_/*A*_sample_) × (Σ[*F*_sample_]/ Σ[*F*_standard_]) × (η^2^_sample_/η^2^_standard_)
where *A* and *F* denote the absorbance and fluorescence intensity, respectively, Σ[*F*] denotes the peak area of the fluorescence spectra, calculated by summation of the fluorescence intensity, and η denotes the refractive index of the solvent (chloroform = 1.444, ethanol = 1.362, water = 1.000). Anthracene (*Ф*_F_ = 0.27 in ethanol) was used as the standard. Extinction coefficients were calculated from the obtained absorbance spectra. This is a standardised method and our values are consistent with the values reported for other profluorescent nitroxides using the same method [17,30,48].

### 3.19. Fluorescence Spectroscopy Measurements: Reduction of FN ***14*** with Sodium Ascorbate 

Sodium ascorbate (200 µM solution in water, 0.5 mL), was added to a solution of FN **14** (2 µM solution in water, 0.5 mL) in a 4-sided quartz cuvette, equipped with a magnetic stirrer bar. The resulting solution (1 µM of FN **14** 1 equiv and 1000 µM sodium ascorbate 1000 equiv) was placed in the fluorescence spectrophotometer, equipped with a magnetic stirrer, and measurements were recorded every minute for 30 mins. The measurement of the blank sample (time = 0 min) was conducted similarly by adding water (0.5 mL) to a solution of FN **14** (2 μM solution in water, 0.5 mL). 

### 3.20. Evaluating the Effect of pH on the Fluorescence Intensity of FM ***17***

Fifteen aqueous solutions (ranging from pH 0 to pH 14) containing the same concentration (500 µM) of FM **17** were prepared. Each respective solution was then analysed via fluorescence spectrophotometery (λ_ex_ = 340 nm), and the total fluorescence area was calculated.

### 3.21. Bacterial Strains and Culture Conditions

*Pseudomonas aeruginosa* ATCC 27853, *Escherichia coli* ATCC 25922, *Staphylococcus aureus* ATCC 29213, and *Enterococcus faecalis* ATCC 19433 were grown routinely in Lysogeny broth (LB) medium with shaking (200 rpm) at 37 °C. Minimum inhibitory concentration (MIC) assays were conducted in Mueller Hinton (MH) medium (OXOID, Thermo Fisher). 

### 3.22. MIC Susceptibility Assays for Compounds ***14***–***19***

The MIC for each fluoroquinolone-based adduct **14**–**19** were determined by the broth microdilution method, in accordance with the 2015 (M07-A10) Clinical and Laboratory Standards Institute (CLSI). In a 96-well plate, twelve two-fold serial dilutions of each compound were prepared to a final volume of 100 µL in MH medium. At the initial time of inoculation, each well was inoculated with 5 × 10^5^ bacterial cells, which had been prepared from fresh overnight cultures in MH. The MIC for a compound was defined as the lowest concentration of an agent that prevented visible bacterial growth after 18 hours of static incubation at 37 °C (MIC values were also confirmed by spectrophotometric analysis at OD_600_). Compounds **14**–**19** were tested between the concentration ranges of 1200 to 0.6 µM. Working solutions of compounds **14**–**19** were prepared in MH medium that had been inoculated with bacteria at approximately 5 × 10^6^ CFU mL^−1^. Negative controls containing DMSO at the highest concentration required to produce a 1200 µM final concentration for compounds **14**–**19** were also prepared and serially diluted (12 dilutions total) in the same method as the antimicrobial agents. MIC values for compounds **14**–**16** were obtained from at least two independent experiments, each consisting of at least three biological replicates and at least two technical replicates of each biological replicate. FNs **14**–**16** and their corresponding FMs **17**–**19** were prepared in DMSO at a concentration of 8 mM (stock solutions) and stored at −20 °C.

### 3.23. Fluorescence Microscopy of Bacterial Cells Treated with FN ***14*** or FM ***17***

Overnight bacterial cultures in LB (10 mL) were concentrated by centrifugation at 3000× *g* for 5 minutes. Cell pellets were washed twice in 10 mL saline (0.9%) and resuspended to approximately 10^9^ CFU mL^−1^ in saline (0.9%). Cell suspensions were treated with FN **14** or FM **17** (150 or 600 µM) for 1.5 hour at 37 °C. Wet mounts (5 µL) of treated cell suspensions were prepared and immediately analyzed by fluorescence microscopy. 

## 4. Conclusions

Several ethyl ester protected fluoroquinolone-nitroxides (**8**–**10**) and their corresponding methoxyamines (**11**–**13**) were prepared using a Buchwald-Hartwig palladium-catalyzed amination coupling in high yield (80–90%) from amino functionalized nitroxides **2**–**4** or methoxyamines **5**–**7**, and the ethyl ester protected fluoroquinolone **1**. Subsequent base mediated deprotection of the ethyl ester protected conjugates **8**–**13** generating the profluorescent FNs **14**–**16** and their corresponding FMs **17**–**19** in high yield (80–90%).

FNs **14**–**16** all exhibited substantially suppressed fluorescence in the presence of the nitroxide moiety (FN **14** 36.7-fold, FN **15** 67.5-fold, and FN **16** 75-fold) when compared to their corresponding FMs **17**–**19**. However, the photophysical properties of FN **14** were determined to be optimal for biological probe applications. We showed that FN **14** permeated several different bacterial species (both Gram-positive and Gram-negative) and fluoresced brightly upon bacterial cell entry exemplifying its potential as an intracellular bacteriological probe. 

The experiments presented here have demonstrated that profluorescent antibiotic nitroxides such as FN **14** possess both desirable fluorescent antibiotic and profluorescent nitroxide probe capabilities. Furthermore, we have successfully produced a novel tool for simultaneously monitoring antibiotic-bacterial interactions and intracellular free radical and redox processes.

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
