# Peer review of "Profluorescent Fluoroquinolone-Nitroxides for Investigating Antibiotic–Bacterial Interactions"

_antibiotics, 2019, doi:10.3390/antibiotics8010019_

Round 1

Reviewer 1 Report

Fairfull-Smith and Totsika report herein the design, synthesis and evaluation of profluorescent fluoroquinolone-nitroxides to study antibiotic-bacterial interactions. The compounds are well characterized and their photophysical properties are properly carried out. Their minimum inhibitory concentration against common Gram-negative and Gram-positive pathogens have been reported as their cellular imaging. I recommend this paper for publication after minor revisions.

1)      Compounds 2-7 should be mentioned under the first arrow in scheme 1. I would also recommend to add 2: RNH2; 3: RNH2, … under the corresponding structures in this scheme.

2)      Compound numbers should be written in bold in the legend of the schemes.

3)      I am not convinced that it is possible to measure a fluorescence quantum yield of 0.0002 using the described method. Could you comment on it?

4)      In table 1, I would recommend to write fluorescence quantum yield instead of quantum yield. It is not clear under which conditions 36.7 is the fluorescence quantum yield ratio of 17/14. It has to be clarified.

5)      P4 line 140: it is not true that FN 14 has improved photophysical properties (especially fluorescence quantum yield) in water compared to chloroform (same quantum yield).

6)      In figure 7, have the absorption spectra been normalized? If not, it should be better to give the values in OD instead of in arbitrary units. Were the emission spectra recorded at the same concentration? It has to be clearly mentioned.

7)      The concentration-dependent fluorescence output of FN14 is not observed in Gram-positive species but is observed in Gram-negative species. Can it be a problem of penetration?

8)      Some procedures have to be modified (for compounds 7 and 8). The aqueous acidic solution is not washed with diethyl ether. The word “washed” in inappropriate. This is rather an extraction. In this case, impurities are extracted. Then, the aqueous solution is basified and the extraction with diethyl ether affords the extraction of the desired compound.

9)      P9, line 336: LC-MS instead of LCMC. More generally, it seems that 2 significant digits are enough for the peak area of LC.

10)   All 1H NMR spectra were recorded at 600MHz but in the supporting information, it is always reported: 1H NMR (CDCl3, 150 MHz). This should be corrected.

Author Response

Reviewer 1

Fairfull-Smith and Totsika report herein the design, synthesis and evaluation of profluorescent fluoroquinolone-nitroxides to study antibiotic-bacterial interactions. The compounds are well characterized and their photophysical properties are properly carried out. Their minimum inhibitory concentration against common Gram-negative and Gram-positive pathogens have been reported as their cellular imaging. I recommend this paper for publication after minor revisions.

1)     Compounds 2-7 should be mentioned under the first arrow in scheme 1. I would also recommend to add 2: RNH2; 3: RNH2, … under the corresponding structures in this scheme.

Scheme 1 has been updated to include these recommendations.

2)     Compound numbers should be written in bold in the legend of the schemes.

Thank you for pointing this out, we have now rectified this in the update version of the manuscript.

3)     I am not convinced that it is possible to measure a fluorescence quantum yield of 0.0002 using the described method. Could you comment on it?

The fluorescence quantum yields were calculated using integrated fluorescence measurements. This is a standardised method and our values are consistent with the values reported for other profluorescent nitroxides using the same method. See Lussini, V, et al DOI: 10.1002/chem.201503393; Ahn, H. et al DOI 10.1021/ja210315x; Allen J, et al https://doi.org/10.1002/ejoc.201601280. This information has now also been added to the manuscript. 

4)     In table 1, I would recommend to write fluorescence quantum yield instead of quantum yield. It is not clear under which conditions 36.7 is the fluorescence quantum yield ratio of 17/14. It has to be clarified.

The word fluorescence has been added to table 1 (column 5), and the superscript of 36.7 has been corrected. 

5)     P4 line 140: it is not true that FN 14 has improved photophysical properties (especially fluorescence quantum yield) in water compared to chloroform (same quantum yield).

We agree that this statement was not correct. This sentence has now been modified in the updated manuscript to read `FN 14and FM 17not only retained their fluorescence in aqueous solution (Figure 2), but FM 17actually produced a higher fluorescence quantum yield, which resulted in an improved suppression ratio in aqueous solution compared to chloroform`

6)     In figure 7, have the absorption spectra been normalized? If not, it should be better to give the values in OD instead of in arbitrary units. Were the emission spectra recorded at the same concentration? It has to be clearly mentioned.

No, the absorption spectra have not been normalized. We have now changed a.u. to O.D. Furthermore, we have added addition information to the caption of Figure 2 `Measured in H2O, λex= 340 nm, and 9 µM for both FN 14and FM 17`.

7)     The concentration-dependent fluorescence output of FN14 is not observed in Gram-positive species but is observed in Gram-negative species. Can it be a problem of penetration?

Our data demonstrate that to achieve fluorescence in the majority of bacterial cells in a treated population you need a higher FN 14 concentration for Gram-negative than for Gram-positive cells. While this could be partly due to reduced penetration of the Gram-negative cell surface, the finding that FN 14 has a lower MIC against E. coli than the Gram-positive E. faecalis would suggest that this phenomenon is multi-factorial and is likely to also be due to the process of fluorescence activation in different bacteria, as indicated in our manuscript. We feel that an investigation into these contributing factors is beyond the scope of this study but would make a valuable follow-up investigation that we are planning to conduct.

8)     Some procedures have to be modified (for compounds 7 and 8). The aqueous acidic solution is not washed with diethyl ether. The word “washed” in inappropriate. This is rather an extraction. In this case, impurities are extracted. Then, the aqueous solution is basified and the extraction with diethyl ether affords the extraction of the desired compound.

We have now changed washed to extracted for both of these procedures.

9)     P9, line 336: LC-MS instead of LCMC. More generally, it seems that 2 significant digits are enough for the peak area of LC.

We have now corrected both of these issues in the updated manuscript.

10)   All1H NMR spectra were recorded at 600MHz but in the supporting information, it is always reported: 1H NMR (CDCl3, 150 MHz). This should be corrected.

This has now been corrected in the updated supplementary information. Thank you for pointing out this error.

Reviewer 2 Report

The manuscript by M. Totsika et al. reports new profluorescent (fluorogenic) fluoroquinolone-nitroxide probes, their spectroscopic characterization and their use in fluorescence microscopy of bacteria. These probes are of significant interest to the community. The experiments are properly described, and the manuscript is well-written. It can be recommended for publication in Antibiotics after addressing the following issues.

1. A general problem associated with the UV-excitable fluorophores is the phototoxicity of UV light and the autofluorescence of living cells. The wavelength used in the fluorescence microscopy experiments described in the paper (365 nm) is commonly used for uncaging of photo-caged molecules, but not for routine live-cell imaging. In view of this, the reader should be informed about possible complications associated with the applications of the FN probes. How intense is the fluorescence of FN probes comparing with the autofluorescence of the bacteria? Is background subtraction needed during the image analysis?  How toxic is the 365 nm excitation to the bacteria? Comments on that should be added to the manuscript.

2. Some of the protonated forms of FM17 shown in Scheme 3 are irrelevant and their existence is not supported by experimental data/literature references. The protonation of N1 is very unlikely to happen, taking into account its electronic conjugation with the carbonyl group. The studies of acid-base equilibria in substituted 4-quinolone carboxylic acids performed by different authors did not reveal such protonated form (see Jelikic̀ et al. Talanta 1992, 39, 665. DOI: 10.1016/0039-9140(92)80078-R). In my opinion, scheme 3 and the supporting discussion on page 5 should be excluded from the manuscript because it will only confuse the reader.

3. Page 4/135: “FM derivatives 18-19 displayed no measurable fluorescence in aqueous solution, indicating water-induced fluorescence quenching”. Is it possible that the low fluorescence was caused by the precipitation of the compounds due to the low solubility in water?

4. Page 5, Figure 3. It would be informative to measure fluorescence of FN14 in the absence of sodium ascorbate (negative control) and also in the presence of LB-medium and freshly prepared E.coli lysates. This would help with interpreting the live cell imaging data.

Some minor issues:

5. The term “small crystals” should be omitted while authors discuss the aggregation of FM17 in water (Page 7/246).  

6. Page 4, Table 1, row “17”, last column. It should be “36.7[b]”, not “36.7[c]”.

Author Response

Reviewer 2:

The manuscript by M. Totsika et al. reports new profluorescent (fluorogenic) fluoroquinolone-nitroxide probes, their spectroscopic characterization and their use in fluorescence microscopy of bacteria. These probes are of significant interest to the community. The experiments are properly described, and the manuscript is well-written. It can be recommended for publication in Antibiotics after addressing the following issues.

1. A general problem associated with the UV-excitable fluorophores is the phototoxicity of UV light and the autofluorescence of living cells. The wavelength used in the fluorescence microscopy experiments described in the paper (365 nm) is commonly used for uncaging of photo-caged molecules, but not for routine live-cell imaging. In view of this, the reader should be informed about possible complications associated with the applications of the FN probes. How intense is the fluorescence of FN probes comparing with the autofluorescence of the bacteria? Is background subtraction needed during the image analysis?  How toxic is the 365 nm excitation to the bacteria? Comments on that should be added to the manuscript.

Thank you for raising these valid points, we have now added an additional paragraph to the manuscript which covers these points. 

Importantly, FN 14exhibited a fluorescence signal which did not require background subtraction or correction for bacteria autofluorescence (bacteria autofluorescence was not detected under these conditions). Furthermore, while the experiments reported here utilized an excitation wavelength of ~365 nm, FN 14was also efficiently excited by a 405 nm laser or a multiphoton laser set at 720 nm (data not shown). Taken together these results demonstrate the utility of FN 14and support its use as a potential live-cell imaging probe.

2. Some of the protonated forms of FM17 shown in Scheme 3 are irrelevant and their existence is not supported by experimental data/literature references. The protonation of N1 is very unlikely to happen, taking into account its electronic conjugation with the carbonyl group. The studies of acid-base equilibria in substituted 4-quinolone carboxylic acids performed by different authors did not reveal such protonated form (see Jelikic̀ et al. Talanta 1992, 39, 665. DOI: 10.1016/0039-9140(92)80078-R). In my opinion, scheme 3 and the supporting discussion on page 5 should be excluded from the manuscript because it will only confuse the reader.

We agree that Scheme 3 is irrelevant and have now removed if from the manuscript. Furthermore, we have also removed the reference to protonation equilibria states from the manuscript text. However, the discussion regarding the pH stability of FN 14 and FN 17 is important for the readers to be aware of as it may influence their intended applications and has thus been retained as is.

3. Page 4/135: “FM derivatives 18-19 displayed no measurable fluorescence in aqueous solution, indicating water-induced fluorescence quenching”. Is it possible that the low fluorescence was caused by the precipitation of the compounds due to the low solubility in water?

We also considered this possibility, however, no precipitation was evident when solutions of these compounds were examined under bright field microscopy with a 40x objective. Furthermore, if precipitation occurred we would have seen a corresponding decrease in the UV absorbance, however, this was not the case.

4. Page 5, Figure 3. It would be informative to measure fluorescence of FN14 in the absence of sodium ascorbate (negative control) and also in the presence of LB-medium and freshly prepared E.coli lysates. This would help with interpreting the live cell imaging data.

Figure 5A shows FN 14 in medium without sodium ascorbate (no measurable fluorescence signal). FN 14 has also been tested in PBS, LB, and MH and does not appear to produce a measurable fluorescence signal via fluorescence microscopy. This information has now been added to the text. `FN 14also emitted no measurable fluorescence in PBS, LB, and MH; data not shown`.

Some minor issues:

5. The term “small crystals” should be omitted while authors discuss the aggregation of FM17 in water (Page 7/246).  

We have now removed the term small crystals from the updated manuscript.

6. Page 4, Table 1, row “17”, last column. It should be “36.7[b]”, not “36.7[c]”.

Thank for point out this error, we have rectified this in the new manuscript.